# Reusable Fe_3_O_4_/SBA15 Nanocomposite as an Efficient Photo-Fenton Catalyst for the Removal of Sulfamethoxazole and Orange II

**DOI:** 10.3390/nano11020533

**Published:** 2021-02-19

**Authors:** Jorge González-Rodríguez, Lucía Fernández, Zulema Vargas-Osorio, Carlos Vázquez-Vázquez, Yolanda Piñeiro, José Rivas, Gumersindo Feijoo, Maria Teresa Moreira

**Affiliations:** 1CRETUS Institute, Department of Chemical Engineering, Universidade de Santiago de Compostela, 15782 Santiago de Compostela, Spain; lucia.fernandezf@gmail.com (L.F.); gumersindo.feijoo@usc.es (G.F.); maite.moreira@usc.es (M.T.M.); 2Laboratory of Magnetism and Nanotechnology, Departments of Physical Chemistry, Faculty of Chemistry and Applied Physics, Faculty of Physics, Universidade de Santiago de Compostela, 15782 Santiago de Compostela, Spain; zulema.vargas@tnuni.sk (Z.V.-O.); carlos.vazquez.vazquez@usc.es (C.V.-V.); y.pineiro.redondo@usc.es (Y.P.); jose.rivas@usc.es (J.R.); 3Centre for Functional and Surface Functionalized Glass, Alexander Dubček University of Trenčín, Študentská 2, 911 50 Trenčín, Slovakia

**Keywords:** magnetic nanoparticles, visible light, photo-Fenton, sulfamethoxazole, Orange II, reuse, reactor

## Abstract

Today, the presence of recalcitrant pollutants in wastewater, such as pharmaceuticals or other organic compounds, is one of the main obstacles to the widespread implementation of water reuse. In this context, the development of innovative processes for their removal becomes necessary to guarantee effluent quality. This work presents the potentiality of magnetic nanoparticles immobilized on SBA-15 mesoporous silica as Fenton and photo-Fenton catalysts under visible light irradiation. The influence of the characteristics of the compounds and nanoparticles on the removal yield was investigated. Once the key aspects of the reaction mechanism were analyzed, to evaluate the feasibility of this process, an azo dye (Orange II) and an antibiotic (sulfamethoxazole) were selected as main target compounds. The concentration of Orange II decreased below the detection limit after two hours of reaction, with mineralization values of 60%. In addition, repeated sequential experiments revealed the recoverability and stability of the nanoparticles in a small-scale reactor. The benchmarking of the obtained results showed a significant improvement of the process using visible light in terms of kinetic performance, comparing the results to the Fenton process conducted at dark. Reusability, yield and easy separation of the catalyst are its main advantages for the industrial application of this process.

## 1. Introduction

Freshwater is one of the most valuable resources of a territory, due to its essential role in life. Its consumption is closely influenced by climatic and anthropogenic factors [1]. Population growth in large cities and the unequal distribution of water, both within territories and over time, put pressure on freshwater resources [2,3,4]. In addition, changes in consumption patterns involve an increased presence of pharmaceuticals, personal care products (PPCPs) and other emerging pollutants in domestic and industrial effluents [5]. Moreover, agriculture causes the release of pesticides that may be present in surface and groundwater by leaching and transport through the soil [6,7].

WWTPs are the main alternative to prevent contamination of freshwater and groundwater. Beyond the removal of organic matter, nutrients and suspended solids, the most recalcitrant compounds present in the water, known as emerging contaminants, can bypass these treatments and reach the effluents. For this reason, some types of tertiary treatments are being studied in recent years in the field of WWTPs to remove these compounds, in order to limit or prevent their release into the environment [8]. Depending on the polarity and persistence of the products, they can leave the WWTPs adsorbed on the biosolids or in the aqueous stream. The diverted fraction can be treated by advanced oxidation processes (AOP), based on the generation of reactive oxygen species (ROS), which are capable of degrading a wide range of compounds through non-specific oxidations [9,10].

In these processes, highly oxidative compounds such as ozone or hydrogen peroxide are used to generate radicals such as·OH to achieve satisfactory removal efficiencies. In the case of ozone-based processes, removal rates above 90% can be achieved for PPCPs [11], the main drawbacks being the incomplete mineralization of the compounds, high energy consumption or limited oxidation of hazardous compounds due to their recalcitrant nature [12]. This removal rates can be improved by combining this technique with other strong oxidants such as H_2_O_2_, UV light or catalysts such as transition metals (Cu(II), Fe(II), ZnO(II), …) or metal oxides (TiO_2_, Fe_2_O_3_, …) [13,14]. Although ozonation is the process applied on a large scale for disinfection, color and odor removal and oxidation of contaminants in drinking water treatment plants, its use is not widespread in WWTPs [13]. In this context, other processes, such as semiconductor-based reactions or the Fenton process, are being investigated for the removal of emerging contaminants.

On the one hand, semiconductor photocatalysis is based on the use of certain metal oxides (commonly TiO_2_ and ZnO) that generate an electro-hole pair when irradiated by a suitable light source, causing the promotion of electrons from the valence band to the conduction band [15]. Oxidation and reduction reactions occur in these pairs involving the adsorption of oxygen on the catalyst surface and hydroxyl anions to produce ROS. To overcome mass transfer limitations, nanostructured materials are commonly used in both semiconductor and Fenton catalysis, since increasing the surface-to-volume ratio increases the reaction rate [16]. Moreover, surface modifications, supporting the nanoparticles onto membranes [17,18], doping with other elements [19,20,21], or targeted design of nanocomposites with magnetic properties have been evaluated to improve performance and enhance the separation stage [22,23]. Moreover, Fenton-based reactions have also generated interest in the field of micropollutant removal because these processes have shown good results even when the target compounds are present in low concentrations [11,24]. Furthermore, the improvement of Fenton processes using light sources up to 600 nm, such as UV or visible light, allows improved reaction rates [16,25]. These types of reactions are based on the use of Fe^2^_+_ as catalyst to enhance the oxidation potential of H_2_O_2_ under acidic conditions (around pH 3) [26] (Equation (1)), to produce mainly · OH (Equation (2)) and ·OOH (Equation (4)). However, other reactive oxygen species (ROS) such as O_2_^−^ and singlet oxygen (^1^O_2_) can be produced during the reactions [27].
(1)$${\mathrm{Fe}}^{2+}+H_{2}O_{2}\to {\mathrm{FeOOH}}^{+}+H^{+},$$
(2)$${\mathrm{FeOOH}}^{+}+H^{+}\to {\mathrm{Fe}}^{3+}+{\mathrm{OH}}^{-}+\;\cdot \mathrm{OH},$$
(3)$${\mathrm{Fe}}^{3+}+H_{2}O_{2}⇌\mathrm{Fe}{\left(O_{2}H\right)}^{2+}+H^{+},$$
(4)$$\mathrm{Fe}{\left(O_{2}H\right)}^{2+}\to {\mathrm{Fe}}^{2+}+\;\cdot \mathrm{OOH},$$

With the use of light sources, the equilibrium reaction for the regeneration of catalyst from Fe^3^_+_ to Fe^2^_+_ is favoured (Equation (3)), positively affecting the kinetics. Further research based on this reaction and applied to the field of wastewater treatment focused on the study and development of innovative catalysts to improve separation, removal rates or shift of reaction conditions towards circumneutral pH [28]. The main drawback of working under these conditions is the precipitation of Fe in the form of insoluble Fe(OH)_2_. Some authors have studied the stabilization of these species using iron complexing agents [29,30] or chemical compounds generated by microorganisms [31], but in these cases, the catalyst is released to the environment. However, the main advantages of Fenton reactions are the operation at standard pressure and temperature conditions, the use of low-risk chemicals and, in the case of homogeneous catalysis, no mass transfer limitations are observed [28]. Nevertheless, it requires the use of iron complexing agents to overcome the precipitation of the species, and separation of the catalyst is not possible, resulting in its continuous loss [32]. Substitution of dissolved iron species by nanostructured catalysts improves separation and reuse in successive removal cycles. Nonetheless, one of the main drawbacks of using heterogeneous catalysts is the decrease in reaction rates due to mass transfer limitations and the presence of active iron only on the surface [28]. At this stage, nanostructured magnetite, a combined oxide containing Fe(II) and Fe(III) species, and its superparamagnetic properties make it ideal for separation by the application of an external magnetic field, as demonstrated in other works [22,33]. However, one of the problems with this material is the formation of aggregates, which reduces it specific surface area negatively affecting the kinetics. Therefore, the enhancement of these NPs by coating with polymers such as polyethyleneimine (PEI) or polyacrylic acid (PAA) generates electrostatic repulsions to avoid agglomeration [33,34].

However, the changes in the surface charge affect the interaction between nanoparticles and target compounds, and for this reason, supporting the nanoparticles on mesoporous silica (SBA-15) increases the specific surface improving the kinetics. Nevertheless, the bottleneck of the Fenton reaction is the regeneration of the catalysts. The incorporation of a light source allows the formation of new reaction routes to regenerate Fe(III), while direct photolysis of hydrogen peroxide occurs. Both changes in the reaction pathways lead to increased reaction rates [28]. Different types of heterogeneous catalysts combining different types of iron oxides or other nanomaterials such as carbon- or cooper-based ones have been used in photo-Fenton reactions. Other authors investigated the use of hybrid materials such as iron oxide/graphene [35], Fe-sand [36] or a combination of different iron oxides or other metal oxides [37,38] to improve the catalytic performance. The main factors studied to evaluate the materials was the reusability of catalysts, their kinetic parameters. However, one of the most important parameters is the feasibility for industrial application, considering the catalyst separation mechanism.

In this research work, the photo-Fenton process was studied, comparing the results obtained with the conventional Fenton process under the same conditions. Different target compounds and types of nanoparticles were analysed, selecting those that showed the best results. In addition, the optimization of the selected variables (catalyst loading, pH and hydrogen peroxide) was investigated using response surface methodology. Finally, the performance of the selected nanoparticles was evaluated under the best conditions in a 2 L reactor for the removal of Orange II (OII) and sulfamethoxazole (SMX), analysing the reuse of the nanoparticles, kinetic performance, degree of mineralisation and effluent toxicity. The use of visible light together with magnetic separation brings this process closer to an industrial application. The replacement of ultraviolet lights with more efficient ones allows reducing operating costs. Moreover, the use of magnetic fields to separate the catalyst improves the process by avoiding the use of a catalyst recovery unit.

## 2. Materials and Methods

### 2.1. Chemicals

Iron (III) chloride hexahydrate (FeCl_3_·6H_2_O, 97% w/w), tetraethyl orthosilicate (TEOS, 98% w/w), polyethyleneimine (PEI, Mw 25000), polyacrylic acid (PAA, Mw 2000), triblock copolymer Pluronic P123 (PEO20-PPO70-PEO20), hydrochloric acid (HCl, 37% w/w), isopropyl alcohol (IPA, 99.7%) and hydrogen peroxide (H_2_O_2_, 30% w/w) were purchased from Sigma Aldrich (St. Louis, MI, USA). Iron (II) sulphate heptahydrate (FeSO_4_·7H_2_O, 99% w/w), ammonium hydroxide (NH_4_OH, 28% w/w), tetramethylammonium hydroxide (TMAOH, ≈10% w/w), orthophosphoric acid (H_3_PO_4_, 85% w/w) and acetonitrile (ACN, HPLC grade) were purchased from Fluka (Buchs, ZU, CH, Switzerland). Sodium hydroxide (NaOH, 95% w/w) was purchased from Panreac Química SLU (Barcelona, CT, Spain). Moreover, the reagents used as target compounds and their details are shown in Appendix A. Milli-Q deionized water was used in all synthesis experiments and was purchased from Merck-Millipore (Darmstadt, HE, Germany). Distilled water or wastewater was used for the preparation of the stock solutions, depending on the type of experiment. In the case of sulfamethoxazole, acetonitrile was used to prepare the initial stock due to the low solubility of this compound in water.

### 2.2. Synthesis and Characterization of Nanoparticles

The synthesis of sterically stabilized black magnetite was conducted following the Massart’s method [39]. For this procedure, a mixture of FeCl_3_·6H_2_O and FeSO_4_·7H_2_O salts with a molar ratio of approximately [Fe^3^_+_]/[Fe^2^_+_] = 1.5 was dissolved into 100 mL of HCl 0.01 M. To promote the formation of black magnetite nanoparticles, 30 mL of NH_4_OH (28% w/w) was added to the mixture and the solution was incubated for 1 h at 60 °C. Once the reaction is over, deionized water was used to wash the synthesized nanoparticles three times and withdrawn by applying an external magnetic field. Finally, a TMAOH solution was used to reach pH 10 and stabilize the nanoparticles in solution [40]. Polymer coating with PEI or PAA was performed in a subsequent step by adding a HCl solution to acidify the medium [41].

On the other hand, the synthesis of magnetite supported by SBA-15 mesoporous silica was based on the Colilla method [42], using a Pluronic P123 (PEO20-PPO70-PEO20) triblock copolymer, HCl/H_3_PO_4_ and TEOS. The obtained product was dried, and several washing cycles were performed with organic solvents to remove the remaining chemicals in order to reduce the adsorption of compounds that could negatively affect the catalytic performance. Furthermore, the synthesis of the immobilized magnetite on mesoporous silica was done by incorporating the SBA-15 matrix into the magnetite synthesis procedure when the iron salts are added [41].

The morphological characterization of the nanoparticles was performed using field emission scanning electron microscopy (Zeiss FE-SEM UltraPlus microscope) with the angle-selective backscatter electron detector (AsB detector) (Zeiss, Oberkochen, BW, Germany). For TEM analysis, a JEOL JEM-1011 transmission electron microscope (Tokyo, 13, Japan) was used. The analysis of the crystalline phases was carried out by X-ray diffraction (XRD) on powder samples with a Philips PW1710 diffractometer (Cu Kα radiation source, λ = 1.54186 Å) (Eindhoven, NB, The Netherlands). Measurements were taken between 15° < 2θ < 80°, increasing by 0.040° and a time per step of 20 s. Pore size distribution and specific surface area were estimated from N_2_ adsorption-desorption isotherms obtained using a Quantachrome Autosorb IQ2 instrument. The ζ−potential was measured by laser-Doppler anemometry in diluted samples (Zetasizer^®^, NanoZS, ZEN 3600, Malvern Instruments, Worcestershire, UK). The nanoparticle concentration of stocks was obtained by thermogravimetric analysis (TGA), and the thermogravimetric curves were recorded using a Perkin Elmer TGA 7 thermobalance (Waltham, MA, USA), operating under nitrogen atmosphere, from room temperature to 850 °C, at a scanning rate of 10 °C min^−1^.

### 2.3. Removal of the Target Compounds

Both the chemical structure and the absorption wavelength of the target compounds are presented in Appendix A. The colored bonds and elements in the semi-structural formulas represent the chromophore molecules (in the case of dyes). The colorant concentrations were measured by visible spectrophotometry in a Powerwave XS2 spectrophotometer purchased from BioTek Instruments (Winooski, VT, USA) using a calibration curve up to 10 mg L^−1^. On the other hand, sulfamethoxazole concentration was quantified by a high-performance liquid chromatography with diode array detector instrument (HPLC-DAD), using a Jasco X-LCTM from Jasco Analítica Spain S.L. (Madrid, MD, Spain) with a Gemini^®^ C₁₈ chromatographic column (3 μm, 110 Å, 150 × 4.6 mm) from Phenomenex Inc. (Torrance, CA, USA).

### 2.4. Photodegradation Experiments

The photodegradation experiments were made in samples of 10 mL and were irradiated using three 11 W PL-G23 fluorescent tubes (4000 K) purchased from Philips (Amsterdam, The Netherlands), with an irradiance of 1.08 W m_−_^2^. Moreover, the experiments made under UV-A (365 nm) Pen-Ray^®^ Lamp 11SC-1L (Analytic Jena, TH, Germany) were irradiated with an intensity of 15.68 W m_−_^2^. On the other hand, the reactor experiments were made in a methacrylate cylindric 5 L small-scale reactor irradiated by three PL-GL23 fluorescent tubes located around the reactor at the same distance used for 10 mL experiments. After degradation experiments, the nanoparticles were separated from the medium by applying an external magnetic field. The paramagnetic properties of the nanocatalysts allow their complete separation from the effluent enabling their reuse in subsequent cycles. Both for visible and ultraviolet light, the values of irradiance were measured using a PCE-UV34 dosimeter (PCE Group Ibérica S.L., Tobarra, CM-ES, Spain), in the range of 290–390 nm. Control experiments of adsorption, photodegradation, stability against pH and stability against hydrogen peroxide were performed to determine the influence of each parameter on the final degradation of the compounds. Thus, the influence of pH medium, incidence of light and hydrogen peroxide concentration was evaluated. In addition, combinations between two factors such as H_2_O_2_ and light were studied. The photodegradation yield (Y) was determined using the following equation (Equation 5):Y(t) = (C_X,i_−C_X,t_)/C_X,i_,(5)
where C is the concentration of the target compound (X) at the starting point (i) or at a given time (t). In addition, for the optimization tests, the experimental data were adjusted according to a pseudo-first-order kinetic model. The linearization of Equation (6) and the expression used to calculate the half-life (Equation (7)) are shown below:ln(C_X,i_/C_X,t_) = kt,(6)
t_1/2,X_ = ln(2)/k,(7)
where t_1/2_ is the half-life of the compound, and k is the kinetic degradation constant for the given conditions.

### 2.5. Wastewater Characterization

To evaluate the effect of the matrix, all the reuse experiments were conducted in a real secondary effluent obtained from a WWTP plant located on Calo-Milladoiro (A Coruña, Spain). This treatment plant has been designed for a capacity of 9000 equivalent inhabitants and consists of a primary treatment for the removal of solids and fats, a biological secondary treatment, and a UV-C treatment for the removal of pathogenic microorganisms. Moreover, the anionic and cationic characterization was performed by ion chromatography, using a Metrohm 861 Advanced Compact IC (Herisau, AR, CH, Switzerland), and the total organic carbon (TOC) was analyzed in Shimadzy TOC 5000 (Kyoto, Japan), both equipped with an autosampler. The pH values were measured using a pH-meter GLP 21 of Crison (Barcelona, CT, ES, Spain). Toxicity assessment was performed by analyzing *Aliivibrio fischeri* bioluminescence in a Microtox^®^ M500 purchased from Modern Water (Guildford, SRY, GB, UK). The chemical characterization of the wastewater is shown in Appendix A.

## 3. Results and Discussion

### 3.1. Nanoparticle Characterization

In this work, different types of magnetite-based nanoparticles were tested for the degradation of different target compounds using Fenton-based methods. Bare and coated magnetite nanoparticles covered with polymers such as polyacrylic acid (PAA) and polyethyleneimine (PEI) were evaluated. These coatings modify the interaction of nanoparticles with the target compounds [40] and improve the kinetic performance [33]. Furthermore, the use of mesoporous silica (SBA-15) as a carrier for bare magnetic nanoparticles was considered to improve magnetic separation, kinetic performance and reusability.

The characterization of free nanoparticles was performed by X-ray diffractometry (XRD), to analyze the crystallographic structure of magnetite. The main peaks found coincide with the characteristic peaks of polycrystalline magnetite (Fe_3_O_4_, JCPDS PDF-2 card #19-0629), indicating that the inverse spinel structure is present in the sample. Furthermore, the absence of other peaks reveals that there are no impurities in the samples. No changes in the peaks were observed upon coating with polyacrylic acid (PAA) or polyethyleneimine (PEI), demonstrating that the coating step does not modify the structure. The complete XRD characterization of this type of nanoparticles was described by Fernández et al. [25]. Appendix A shows the Fe_3_O_4_@PAA/SBA15 nanoparticles as prepared and after 1, 5 and 10 reuse cycles. As observed in these nanoparticles, the characteristic peaks of magnetite appear in the XRD pattern, but a higher signal/noise ratio was obtained in the measurements due to the presence of SBA15 mesoporous silica matrix.

The size distribution of unsupported nanoparticles was measured by TEM images, calculating the mean and standard deviation of Feret diameters for N > 100 measurements. These diameters range from 7.6 ± 2.7 nm for PEI-coated nanoparticles to 10.9 ± 3.0 nm for PAA-coated ones. The bare magnetite nanoparticles have a diameter of 8.2 ± 3.8 nm. In addition, the zeta potential of the NPs was determined at a near-to-optimum pH for Fenton reactions (pH 3), resulting in −11.452 mV for the Fe_3_O_4_ nanoparticles and −4.531 and 29.942 mV respectively for the PAA and PEI-coated magnetite. The zeta potential values of Fe_3_O_4_@PAA/SBA-15 as a function of pH are shown in Figure 1. Considering the results obtained, the point of zero charge (PZC) of these nanoparticles was reached around pH 2, reaching a value of −4.27 ± 0.21 mV for the working point (pH 3).

On the other hand, the morphological characterization of the nanoparticles immobilised on SBA15 mesoporous silica (SBA15/Fe_3_O_4_@PAA and SBA15/Fe_3_O_4_@PEI) was performed using SEM and TEM images. Figure 2 shows the TEM and SEM images for Fe_3_O_4_@PAA/SBA15 nanoparticles, being the complete characterization of the rest of the nanoparticles in Fernández et al. [25].

In this case, magnetite nanoparticles are isolated or forming aggregates on the surface of the 2D hexagonal structure of SBA-15. Although the deposition of magnetite inside the pores cannot be clearly observed by microscopy, the presence of magnetite nanoparticles and aggregates is expected also inside the pores of SBA15 considering the synthesis procedure. Adsorption/desorption isotherms with N_2_ were performed to obtain the surface and volumetric parameters. The adsorption isotherms and pore-size distribution of the SBA15/Fe_3_O_4_@PAA nanoparticles are depicted in Figure 3.

For SBA15/Fe_3_O_4_@PAA, the specific surface was 220.3 m^2^ g^−1^, and the pore volume was 0.810 cm^3^ g^−1^, with an average size of 7.588 nm. On the other hand, for SBA15/Fe_3_O_4_@PEI, the synthesis was conducted using a PEI with Mw of 25,000. In this case, the specific surface was 157.3 m^2^ g^−1^, and the pore volume was 0.44 cm^3^ g^−1^.

### 3.2. Potential of Nanocomposites for Dye Removal

The non-specific oxidation mechanisms of the Fenton and photo-Fenton processes can oxidize a wide range of chemicals, regardless of the structure of the molecule. In the first step of the evaluation of the oxidative capacity of nanocatalysts, anionic and cationic dyes were used as model compounds: copper phthalocyanine (Cu(II)Pc), methyl green (MG), eriochrome black T (EBT), reactive blue 19 (RB19) and Orange II (OII), whose characteristics and structures are shown in Appendix A. Their use allows a rapid assessment of the concentration by spectrophotometric measurements in order to determine the kinetic data. In this section, experiments have been performed using SBA15/Fe_3_O_4_@PAA nanoparticles, and considering similar conditions to those used by other authors (pH 3, 300 mg L^−1^ of H_2_O_2_) [28,33] for a dye concentration of 10 mg L^−1^ with white light irradiation. The results obtained are shown in Figure 4.

The adsorption control experiments were performed using the same concentration of nanoparticles without the use of light or hydrogen peroxide, and the final concentration was measured after 2 h. For the case of photo-Fenton process, a combined control of hydrogen peroxide and light was conducted, and for the Fenton reaction, this control was performed under dark conditions. In terms of dye removal, OII is the most recalcitrant compound, since it presents the lowest percentage of degradation. To study the contributions of the independent factors in the degradation due to each factor, stability tests were performed. The stability control for pH indicates that no degradation occurred in the tests performed at acidic conditions. Adsorption values were fairly different between the target compounds according to their anionic or cationic nature. The planar structure of Cu(II)Pc and its low solubility of phthalocyanines in water lead to complete adsorption on the catalyst with the molecules stacked on top of each other. On the other hand, the zeta potential (ζ) of SBA15/Fe_3_O_4_@PAA at pH 3 is −1.468 mV, which explains that the negative surface charge causes electrostatic repulsions with the anionic dyes (OII and RB). The enhanced adsorption of EBT could occur due to the protonation of the azo groups, reducing the partial negative charge of the molecule and aiding adsorption onto the catalyst. The cationic nature of MG was responsible for the partial adsorption of the dye: 59%. The catalyst performance was strongly affected by the interaction between nanoparticles and dyes as a function of the electrostatic charges of the molecules. Compounds that reached high adsorption values such as MG or EBT were similarly removed in the Fenton and photo-Fenton processes, rendering low light irradiation efficiency. This phenomenon was possibly due to the fact that their adsorption on the catalyst surface caused a shielding of the light irradiation and consequently reduced the removal yield.

Degradation due to hydrogen peroxide in an acidic medium (pH 3) occurred to a greater extent for EBT and RB, indicating that they are less recalcitrant compounds than MG and OII, whose degradation was not complete at the end of the experiment. Moreover, as observed in Figure 4, the dyes do not act as photosensitizers because the degradation percentages in the absence of nanoparticles are not significant (especially in the cases of Cu(II)Ph, MG and OII). The kinetic constants obtained in the degradation of MG and OII were 0.889–1.322 h^−1^ and 0.543–1.141 h^−1^ for the Fenton and photo-Fenton processes, respectively. In all cases, the correlation coefficients were greater than 0.99, demonstrating the good fit of the pseudo-first model. The differences between Fenton and photo-Fenton kinetic constants were 49% for photo-Fenton and 15% for Fenton, showing significantly lower OII removal rates in the latter case. Considering that the OII dye is the most recalcitrant dye, as long as this compound is removed by the nanocomposite, the rest of the dyes should be removed.

### 3.3. Screening of Different Types of Nanoparticles

Different types of magnetite-based catalysts were tested for their performance in Orange II dye removal. The influence of magnetite coatings using TMAOH-stabilized Fe_3_O_4_, Fe_3_O_4_@PAA and Fe_3_O_4_@PEI and the immobilization on mesoporous silica on Fe_3_O_4_@PAA/SBA15 and Fe_3_O_4_@PEI/SBA15 nanocomposites were evaluated on the basis of the use of different types of catalyst with equivalent magnetite loading. A concentration of 200 mg L^−1^ of Fe_3_O_4_ and H_2_O_2_ was used for the degradation of 10 mg L^−1^ of Orange II at pH 3. The degradation results are shown in Figure 5.

Under these conditions, Fe_3_O_4_@PAA/SBA15 provides the best results for OII degradation, achieving complete removal of the dye for Fenton and photo-Fenton processes. However, the estimation of the kinetic constant showed that the reaction rate under irradiation was approximately 5 times higher, with a half-life of 15 min in the case of photo-Fenton, showing a R^2^ value greater than 0.98 in all cases. The high positive value of the zeta potential for Fe_3_O_4_@PEI nanoparticles at pH 3 causes a remarkable adsorption of the OII dye. In order to quantify the adsorption on the NPs catalyst, the pH of the medium was changed to 7 at the end of the degradation experiments to desorb the remaining dye. The difference between the initial and final concentration was less than 10%, denoting the predominant contribution of adsorption. This phenomenon could be due to the fact that the high interaction between the compound and catalysts caused surface saturation, preventing radical formation and decreasing reaction rates. On the other hand, the uncoated nanoparticles showed better results than those coated with PAA. However, their stability in aqueous media is lower, and they precipitate out of water to form agglomerates, which could hinder their industrial application. Although bare magnetite NPs showed a high zeta potential value in absolute terms compared to those coated with PAA or PEI, electrostatic repulsions were not sufficient to prevent agglomeration. The low size of the NPs causes them to move closer to each other, and magnetic interaction plays an important role in this phenomenon. The agglomeration also causes the reduction of the oxidative efficiency due to the decrease of the active surface area. In view of the results, Fe_3_O_4_@PAA/SBA15 was selected as catalysts for the next steps and used for the optimization of the operating conditions, such as the catalyst loading or the hydrogen peroxide concentration.

### 3.4. Optimization of Photo-Fenton Parameters

#### Selection of Variables for Optimization

The optimization of the reaction conditions took into account not only the influence of operational parameters on the removal of dye but also the selection of conditions that would ensure a balance between efficiency and reduction of operational costs, with a view on prospective application. To determine the influence of variables on OII removal, response surface methodology (RSM) was applied according to a Box-Behnken design, studying the catalyst loading between 200 and 500 mg L^−1^, H_2_O_2_ concentration between 100 and 300 mg L^−1^ and pH in the range of 3 to 7. Considering that a pH value lower than 3 could cause the solubilization of magnetite [28], this point was selected as the lower limit of the study and pH 7 as the upper limit, this value corresponding to the pH of the effluent (pH 6.5–7.5). In addition, a limit of 300 mg L^−1^ of hydrogen peroxide concentration was selected because an excessive concentration of H_2_O_2_ can quench the process by reacting with the radicals formed [21]. Considering the non-dependence of the kinetics with the dye concentration (pseudo-first order model), an initial dye concentration of 10 mg L^−1^ was used for all experiments.

Analysing the results of the effects for each variable and considering the error calculated by the replicates at the central point, four significant effects were obtained for *p* < 0.05, indicating their influence on the kinetic constant with a probability of more than 95%. These factors were the pH (both in linear and quadratic terms), the H_2_O_2_ concentration and the interaction factor between these two variables. The system showed a similar performance compared to the work reported by other authors [43], in which there was a minor influence of the catalyst concentration but a high dependence on pH, with high reaction rates for lower pH. The Pareto chart is shown in Figure 6, in which the less influential variables were discarded to recalculate the error bar.

The regression coefficient R^2^ provided a value of 0.976, indicating that the model adequately fitted the system performance. The results obtained for the coefficients for all values are shown in Appendix A. The influence obtained for pH was a convex polynomial, parameterized by a second-degree equation, with a minimum at pH 6.5. Throughout the interval studied, the optimum pH was 3. In addition, the influence of H_2_O_2_ concentration was linear, with a maximum reaction rate at a load of 300 mg L^−1^, and the influence of the catalyst concentration was not significant according to its *p* value (Figure 7).

Taking into account the obtained results and considering that the influence of the catalyst load is not statistically significant, the selected concentration was 200 mg L^−1^, which allows a lower cost related to the use of nanoparticles. In addition, the optimal concentration of peroxide in this interval was 300 mg L^−1^.

Analyzing the results, the optimum point is outside the evaluated range, but considering that the increase of H_2_O_2_ concentration leads to an increase of chemical consumption, and its influence on the reaction rate is not high, this value was selected for the following experimental steps. The best results were obtained for pH 3, which is an operational limit considering that a lower pH would entail the solubilization of magnetite. Under these conditions, the theoretical value of the kinetic constant was 4.014 h^−1^. However, as this is an extreme point in the Box–Behnken design, this value was not obtained experimentally but calculated by extrapolation. For this reason, further experiments were performed for a catalyst load of 200 mg L^−1^, a hydrogen peroxide concentration of 300 mg L^−1^ and at pH 3.

### 3.5. Proof-Of-Concept of Nanoparticles Reuse

#### 3.5.1. Orange II Removal

Once the best case was established, a performance analysis was carried out over ten reuse cycles to examine the stability of the catalyst in a real wastewater matrix. The conditions defined in the optimization study were used: catalyst loading of 200 mg L^−1^ and H_2_O_2_ concentration of 300 mg L^−1^ at pH 3 under visible light irradiation. The experiments were carried out for 2 h, sampling every 15 min in the first hour and every 30 min thereafter.

The removal of OII below the detection limit was achieved in all cases within 2 h. TOC decrease was evidenced between 40 and 60% during the first nine cycles and fell below these values in the last cycle. The TOC concentration after treatment may be indicative of the mineralization of the organic matter present in the wastewater. Since the degradation of OII and SMX is non-specific and the reaction pathways are branched, the TOC values provide global information on the extent of degradation of the target compounds after treatment. Considering the molecular composition of OII and SMX, the expected reaction products after complete mineralization are CO_2_, H_2_O, NH_4__+_, NO_3_^−^, SO_4_^−^ and mineral acids [44].

The average kinetic constant was 1.5 h^−1^, being approximately 50% lower than the one calculated theoretically from the experimental design. One of the causes of this decline could be the presence of cations such as Li_+_, Na_+_, NH_4__+_, K_+_, Mg^2+^ or Ca^2+^ in the wastewater matrix, which could interact with the negatively charged SBA15/Fe_3_O_4_@PAA surface, thus blocking the access of the dye to the surface of the nanocatalyst and inhibiting degradation. Apart from this, certain anions such as SO_4_^2−^, Cl^−^ or CO_3_^2−^ may react with the ROS, acting as oxygen scavengers [45].

In addition, control tests were performed using the same reaction time, obtaining 2.5% OII removal by photolysis, 2.2% degradation by H_2_O_2_ and 7.6% by the combined effect of light and H_2_O_2_. Adsorption controls were also performed, with values of 17.2% in the first cycle and 1.1% in the second. These results suggest that most of the adsorption occurs when using an unused catalyst, and in subsequent uses, the surface of the catalyst became saturated with organic matter, showing a lower adsorption capacity for OII. Moreover, no dissolved iron was detected in the effluents of the cycles, verifying that there is no contribution from homogeneous photo-Fenton reactions. The results of the cycles are shown in Figure 8.

Furthermore, the MicroTox^®^ tests indicated that the decrease in toxicity is largely due to chronic effects (15 and 30 min), showing minor effects due to acute toxicity (5 min) (Appendix A). According to the results, the reduction in toxicity is not affected by the reuse of the catalysts. In addition, these results showed that no intermediate compounds are produced that may increase the toxicity of the effluent.

Moreover, the reuse tests were carried out for the Fenton mechanism (without light), in order to observe the influence of the irradiation on degradation. For these experiments, the same experimental conditions were maintained, only suppressing the light irradiation. For the experiments conducted at pH 5 and 7, there was no degradation of OII, regardless of the catalyst load and the hydrogen peroxide concentration. For this reason, the optimization was performed at pH 3. The obtained kinetic constants for these experiments are shown in Appendix A. In addition, the same conditions were selected to perform all the reuse tests, to determine the influence of light on the process.

As in the previous experiments, a real wastewater matrix was used for these cycles, maintaining an initial concentration of 10 mg L^−1^ of OII. The kinetic constants obtained were, on average, four times lower than those obtained for the photo-Fenton experiments under the same conditions. On the other hand, their value was maintained throughout the cycles with slight variations, indicating that the reuse of the catalyst does not affect its performance (Appendix A).

#### 3.5.2. Sulfamethoxazole Removal

After studying the performance of the catalyst against OII, reuse tests for SMX degradation were carried out under the same conditions. The removal of SMX remained similar throughout the cycles in values around 80–90%. The TOC removal values were between 5–45%, and these values were lower than those obtained for OII, possibly because the initial value of this parameter was higher (Figure 9).

On the other hand, the experiments were performed under the same conditions avoiding the use of light, in order to quantify the contribution of irradiation to the process (Appendix A). In this case, the kinetic constant reached values close to 0.1 h^−1^, 50% lower than those obtained for OII removal. As in the previous cases, a decrease in the reaction rate was observed in the first cycles and a stabilization of the value thereafter. This performance could be explained because the generation of reaction intermediates could affect the catalyst, by adsorption on its surface and reducing its active area as the different cycles are carried out. It should be noted that the catalyst is not regenerated or cleaned, only separated with a magnet and resuspended in the new matrix. Complete SMX removal was not reached in any experiment after 4 h, and TOC removal reached a maximum value of 40%. Following the trends of the other experiments, the kinetic constant remains stable after the first cycles.

### 3.6. Pilot Scale Reuse Cycles

#### 3.6.1. Orange II Removal

The removal of the dye was conducted in a 5 L reactor (with a useful volume of 2 L) irradiated by three 11 W fluorescent lamps. Mechanical stirring was used to ensure proper mixing of the reaction volume, to avoid the interactions between magnetic fields and nanoparticles. The magnetic separation was performed by introducing a magnetic bar inside a glass tube in the volume of the reactor [46]. The results for the OII removal are shown in Figure 10.

After two hours of reaction, the OII dye was completely removed and the TOC concentration was partially reduced at a large extent, with removal values higher than those of laboratory scale experiments (up to 75%). Moreover, the toxicity values, related to the formation of toxic degradation intermediates, are lower than those obtained for the initial effluent (Table 1).

#### 3.6.2. Sulfamethoxazole Removal

Finally, the nanocatalyst was reused to check the elimination of SMX. The kinetic constants of SMX removal are half of those achieved for OII under the same conditions. The TOC removal percentages reached values above 60% and the degradations of the target compound were, except in the first cycle, above 75% after five hours of reaction (Figure 8). The degradation values were in the ranges of the obtained by other authors both for photo-Fenton processes [44,47] and semiconductor photocatalysis [48]. The differences on the values could be originated because the use of a real wastewater matrix can interact to radical formation, diminishing the efficiency of the process. Moreover, the use of the 2 L reactor causes a reduction in kinetic performance possibly due to the inefficient irradiation of samples, as evidenced when comparing Figure 9 and Figure 11.

On the other hand, the main advantage of these nanocatalysts compared to the conventional ones is the easy separation of the nanoparticles from the effluent, improving their industrial application.

The changes in the kinetic constant follow the same trends observed at laboratory scale. The variation of the kinetic constant values at the initial cycles could be explained by the adsorption of by-products on the catalyst surface. After the third cycle, equilibrium was reached and the differences between these values are reduced. Moreover, the differences between the kinetic constants of OII and SMX could be related to the electromagnetic interactions between the catalysts and the target compounds. In the case of OII, the group N = N has a partial positive charge mainly due to the delocalization of the charge in the carbon rings, and it has a positive charge in the hydrogen of -OH group. These positive charges increase the affinity for the negatively charged surface of the catalyst. On the other hand, the most positively charged element in SMX is sulfur, but this charge is balanced out by the negative charges of the oxygen, and the other positive charge (O of the ring) is counteracted by the N, decreasing the interaction between this compound and the catalyst.

Considering the degradation pathways, advanced oxidation processes in Fenton-type processes follow non-specific oxidation pathways. In the case of sulfamethoxazole, previous studies reported that the degradation pathway in Fenton reactions could start by the oxidation of the amino group on the benzene ring, cleavage of the bond between the sulfur and the benzene ring, oxidation of the amino group or attack of the ·OH on the benzene ring and the isoxazole double bond. Thus, the degradation pathways are not unique and the formation of a wide range of intermediates takes place [44,49,50]. In addition, sulfamethoxazole and Orange II were used as model compounds due to their recalcitrant characteristics. Based on the degradation results, the application of this process to real wastewater with a wide range of compounds is expected to be a viable alternative.

### 3.7. Nanoparticle Stability after Reuse

In order to monitor the stability of Fe_3_O_4_@PAA/SBA15, TEM (Figure 2A) and SEM (Figure 2B), images were taken at the beginning of the experiment and after 10 reuse cycles (Figure 2C,D). No changes were observed in the structure of SBA15, and the magnetite remains on the surface; in addition, the 2D hexagonal structure of the mesoporous silica is still observed. In addition, the stability of similar type of nanoparticles has been demonstrated by other works [33].

On the other hand, XRD measurements were made initially and after the first, fifth and tenth cycles (Appendix A) to observe changes in morphology and structure. When comparing the results, no differences were observed in the morphology of the catalyst or in the wide-angle XRD. However, the diffraction peak at 0.7° 2θ disappears in the low-angle XRD analysis for the reused cycles. Both the TEM/SEM images and the XRD characterization of catalyst before and after the treatment showed that there is no evidence of inactivation of catalyst or structure changes in the nanoparticles. Moreover, the kinetic rates remain stable as early as the second cycle, suffering only slight changes compared to the initial values.

## 4. Conclusions

Advanced oxidation processes based on Fenton and photo-Fenton reactions are promising technologies in the field of removing emerging contaminants in wastewater. In this work, the use of coated magnetic nanoparticles or their combination in nanocomposites supported on mesoporous silica for the removal of different dyes and sulfamethoxazole was investigated. According to the results, the best performance was obtained for magnetite nanoparticles supported on SBA15 and coated with polyacrylic acid. The response surface methodology was applied to identify the influence of pH as the most influential variable on the degradation of the target compounds, followed by H_2_O_2_. Surprisingly, for the range studied, the influence of catalyst loading was not significant. These optimum conditions were applied in the stability tests. Reusability of the catalyst was demonstrated after 10 cycles of reuse, showing low variability in the kinetics. Moreover, in all these experiments, an external magnetic field was used to separate the magnetite nanoparticles from the treated effluent, which points to this technology as a suitable process to apply in wastewater treatment.

## Figures and Tables

**Figure 1 nanomaterials-11-00533-f001:**
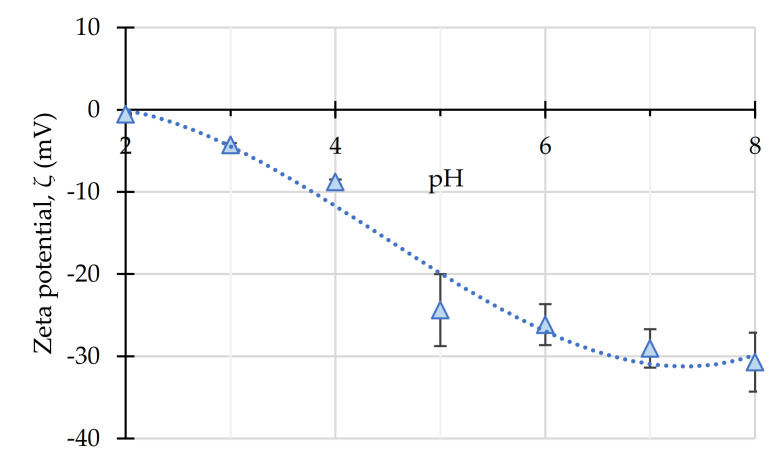
Zeta potential of SBA15/Fe_3_O_4_@PAA nanoparticles for a pH range between 2 and 8. The blue dotted line is a guide to the eye.

**Figure 2 nanomaterials-11-00533-f002:**
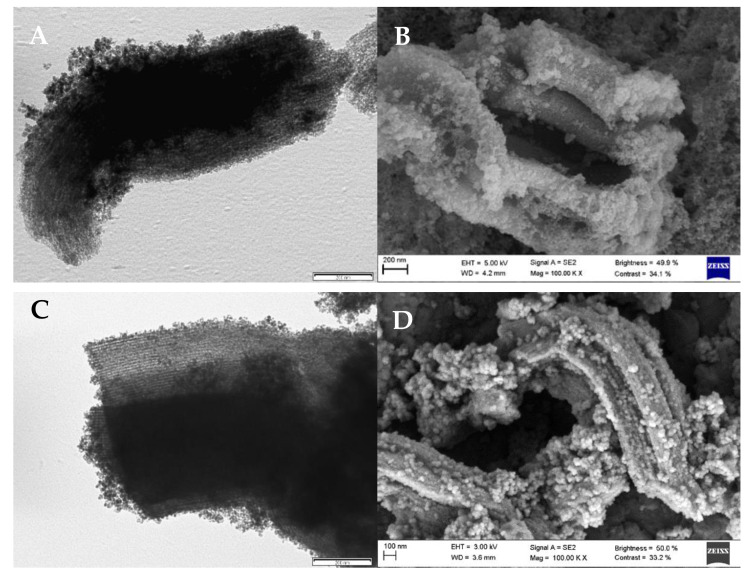
(**A**) TEM and (**B**) SEM images of Fe_3_O_4_@PAA/SBA15 nanoparticles as prepared and (**C**) TEM and (**D**) SEM images after ten reuse cycles.

**Figure 3 nanomaterials-11-00533-f003:**
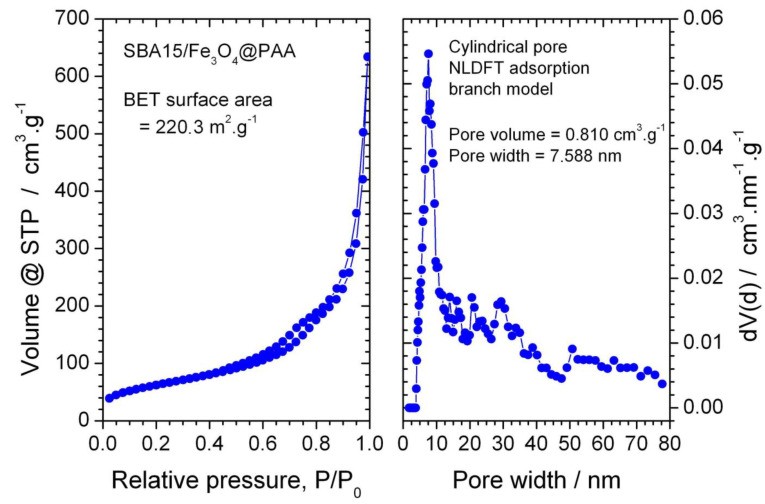
Adsorption isotherms and pore-size distribution of SBA15/Fe_3_O_4_@PAA nanoparticles.

**Figure 4 nanomaterials-11-00533-f004:**
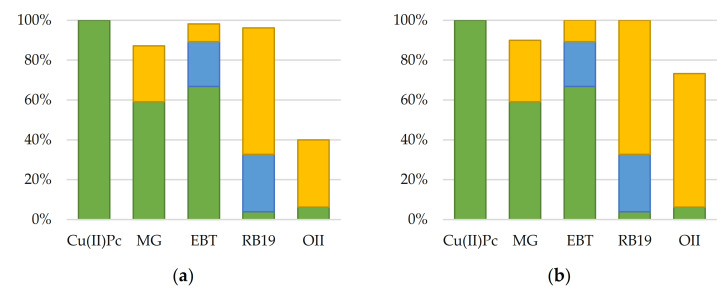
Comparison of Fenton (**a**) and photo-Fenton (**b**) potentials for dye removal. [Cat] = 200 mg L^−1^; [H_2_O_2_] = 300 mg L^−1^; [MC]_0_ = 10 mg L^−1^. Adsorption control (green), light/dark + H_2_O_2_ control (blue) and degradation (yellow).

**Figure 5 nanomaterials-11-00533-f005:**
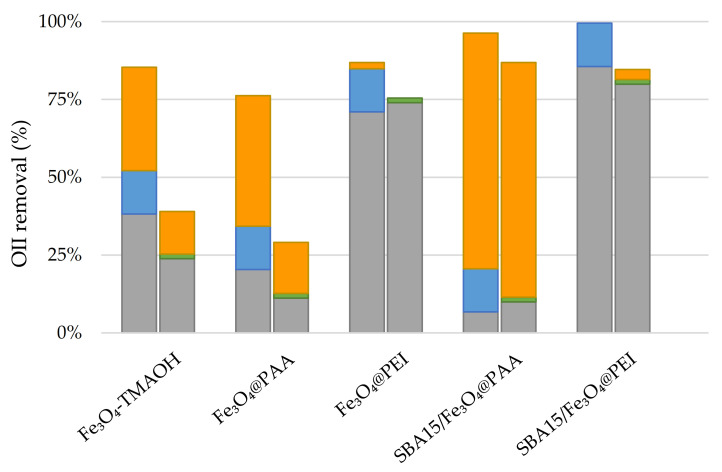
Comparison of catalyst performance after 3 h for adsorption (grey), H_2_O_2_ (green), light + H_2_O_2_ (blue) and Fenton (orange, right) and photo-Fenton (orange, left) contribution, for [Fe_3_O_4_] = 200 mg L^−1^, [OII] = 10 mg L^−1^.

**Figure 6 nanomaterials-11-00533-f006:**
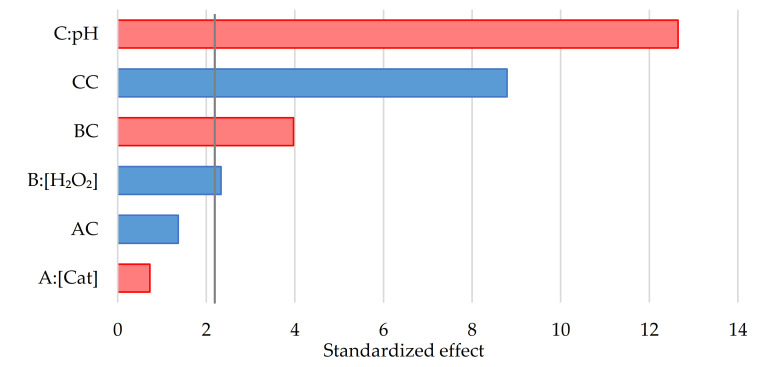
Pareto chart of the standardized negative (red) and positive (blue) effects for the photo-Fenton process. Significance limit marked with a vertical grey line (value 2.25).

**Figure 7 nanomaterials-11-00533-f007:**
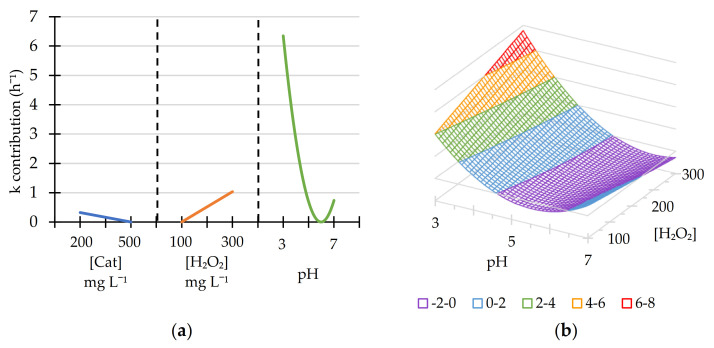
Contribution of the variables to the kinetic constant (**a**) and the response surface for k values (**b**). [Cat] = 200 mg L^−1^.

**Figure 8 nanomaterials-11-00533-f008:**
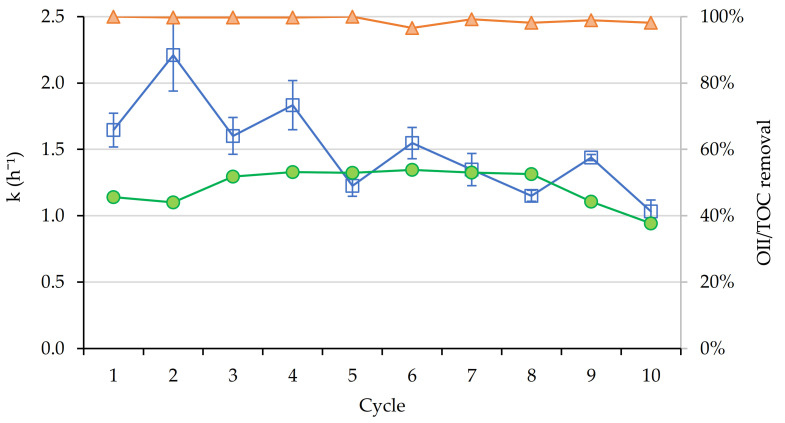
Kinetic constants (h^−1^) (blue), TOC (green) and OII removal (orange) for lab-scale photo-Fenton experiments (visible light).

**Figure 9 nanomaterials-11-00533-f009:**
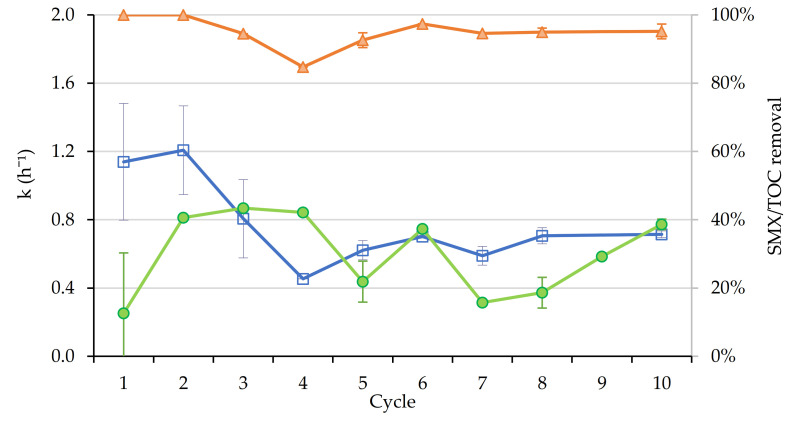
Kinetic constants (h^−1^) (blue), TOC (green) and SMX removal (orange) for lab-scale photo-Fenton experiments (visible light).

**Figure 10 nanomaterials-11-00533-f010:**
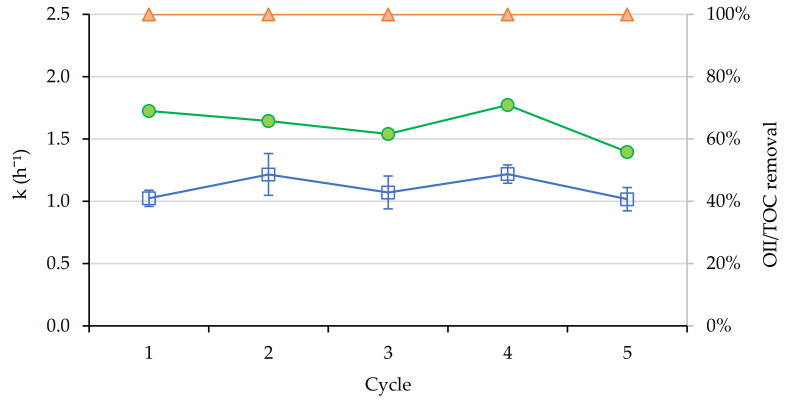
Kinetic constants (h^−1^) (blue), TOC (green) and OII removal (orange) in two liters reactor for photo-Fenton experiments (visible light).

**Figure 11 nanomaterials-11-00533-f011:**
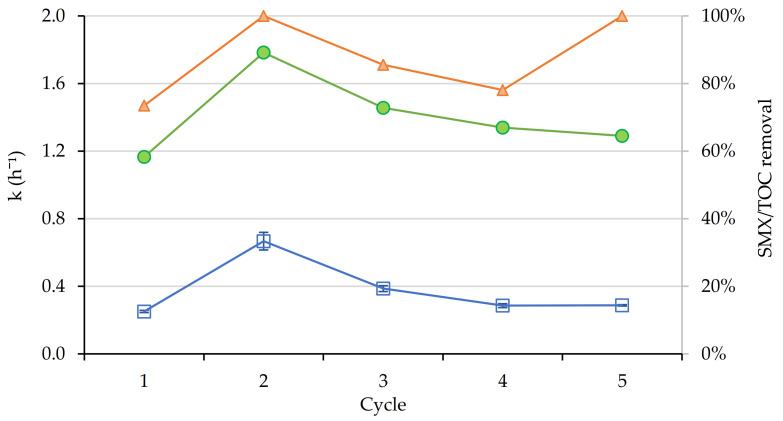
Kinetic constants (h^−1^) (blue), TOC (green) and SMX removal (orange) in two liters reactor for photo-Fenton experiments (visible light).

**Table 1 nanomaterials-11-00533-t001:** MicroTox^®^ results of different treated effluents for small-scale OII removal.

Sample	EC_50,5_	EC_50,15_	EC_50,30_
Initial toxicity	9	5	4
Effluent cycle 1	12	n/d	16
Effluent cycle 2	19	15	21
Effluent cycle 3	14	22	16

## Data Availability

The study did not report any data.

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
