# Peer review of "Reusable Fe3O4/SBA15 Nanocomposite as an Efficient Photo-Fenton Catalyst for the Removal of Sulfamethoxazole and Orange II"

_nanomaterials, 2021, doi:10.3390/nano11020533_

Round 1

Reviewer 1 Report

In this manuscript, the authors present results pertaining to catalytic degradation of organic species in water using a high-surface-area magnetic Fe/Si material via Fenton reactions. Recalcitrant organic pollutants are among the major classes of emerging contaminants in wastewater, particularly in the context of pharmaceutical agents that can concentrate in the effluent from treatment plants. Traditional water treatment methods do not remove these compounds, so many researchers are exploring the use of catalysis to degrade the species into (hopefully) harmless products. Advanced oxidation processes of various types have been reported, and many of the effective approaches rely on addition of H2O2 to seed Fenton (or photo-Fenton) reactions. Visible-light photocatalysis is particularly of interest due to the potential to use sunlight as the driving force, which limits the palette of available catalysts since the majority rely on UV irradiation. The authors also utilize magnetic field to capture the particles after treatment for recycling. Each of these aspects has been reported previously, but there is some novelty in the current study with regard to the specific materials synthesized and the range of process conditions sampled. The manuscript is well-written, the analysis is thorough, and the results are compelling overall. This work is suitable for publication, pending revision to address the following minor issues:

- The authors actually undersell their work in the abstract. They have probed the degradation of various dyes, but the abstract only mentions Orange II. This was the dye that exhibited the most degradation in their experiments, but they actually sampled a range of dyes, and from these, they were able to decipher key aspects of the reaction mechanism (based on the disparate electrostatic charges). It is worth highlighting this point in the abstract. On the other hand, it is also important to mention since this particular material may not be suitable for cationic dyes.

- The introduction would benefit from additional context on other visible-light photocatalysts for water treatment applications, perhaps most notably doped titanium dioxide (e.g. Adv. Sust. Sys. 1 (2017) 1600041). Also, the citations miss a recent report on Fe(OH)2 nanorod/PDA visible-light photocatalysis (Adv. Funct. Mater. 27 (2017) 1700251).

- The authors perform TOC analysis to quantify the degree of mineralization, which is indeed useful to know. It would also be useful to know the nature of the reaction byproducts for those molecules that are not fully mineralized. This may be beyond the scope of the current study, but it would still be beneficial to discuss potential pathways.

Reviewer 2 Report

The paper is too preliminary for publication. It should be rejected. New experiments should be done before it can be re-considered as a new paper.

  1. Figure S1. The signal/noise level is so low. The data should be smoothed by 5-point-smoothing function of Origin.
  2. BET surface areas, adsorption-desorption isotherms, and pore-size distribution data should be provided.
  3. Please be reminded that the materials characterization part is also important for a paper submitted to Nanomaterials. At least the adsorption-desorption isotherms, pore-size distribution data, and TEM/SEM images should be moved to the main text. Currently, the focus of the paper is only regular testing. That is not enough.
  4. The reaction mechanism is not clear at all. I’ll have to suggest rejection if the authors bring back the paper without telling us about the reaction mechanism.
  5. There are errors in the references section. For instance, be careful about the usage of subscript in ref. 37.
  6. The authors only tested the TOC values, without telling us about the nature and composition of reaction products.
  7. Is the catalyst magnetic? What is the advantage of the catalyst?

Reviewer 3 Report

This work focuses on the potentiality of magnetic nanoparticles immobilized on SBA-15 mesoporous silica as Fenton and photo-Fenton catalysts under visible light irradiation. In order to evaluate the feasibility of this process, an azo dye (Orange II) and an antibiotic (sulfamethoxazole) were selected as target compounds.

 In order to evaluate the feasibility of this process, an azo dye (Orange II) and an antibiotic (sulfamethoxazole) were selected as target compounds. The concentration of Orange II decreased below the detection limit after two hours of reaction, with mineralization values of 60%. In addition, repeated sequential experiments revealed the recoverability and stability of the nanoparticles in a small-scale reactor.

This is indeed a good work. It is well written and can be published after minor revisions.

1. I feel that a few more references could be added.

For example, the authors state in line 58: "..This removal rates can be improved by combining this technique with other strong oxidants such as Hâ‚‚Oâ‚‚, UV light or catalysts such as transition metals (Cu(II), Fe(II), ZnO(II), …) or metal oxides (TiOâ‚‚, Feâ‚‚O₃, …).." a reference is missing there.

Moreover, in lines 69, 74 and 80, a reference could be added.

Please check the introduction part and add new references at several points

2. The authors should discuss further about the kinetics behavior in Figs. 6-8: We can see that the Kinetic constants are decreasing with the number of cycles, and then they ares increasing again. Please explain and discuss further.

3. A few typos and syntax errors should be corrected.

Round 2

Reviewer 2 Report

accept